# Computational Explorations of Total Variation Distance

**Arnab Bhattacharyya**[*]
University of Warwick

**Sutanu Gayen**
IIT Kanpur

**Kuldeep S. Meel**
University of Toronto
Georgia Institute of Technology

**Dimitrios Myrisiotis**
CNRS@CREATE LTD.

**A. Pavan**
Iowa State University

**N. V. Vinodchandran**
University of Nebraska-Lincoln

## Abstract

We investigate some previously unexplored (or underexplored) computational aspects of total variation (TV) distance. First, we give a simple deterministic polynomial-time algorithm for checking equivalence between mixtures of product distributions, over arbitrary alphabets. This corresponds to a special case, whereby the TV distance between the two distributions is zero. Second, we prove that unless NP $\subseteq$ RP it is impossible to efficiently estimate the TV distance between arbitrary Ising models, even in a bounded-error randomized setting.

## 1 Introduction

The total variation (TV) distance between distributions $P$ and $Q$ over a common sample space $D$, denoted by $d_{\mathrm{TV}}(P, Q)$, is defined as

$$d_{\mathrm{TV}}(P, Q) := \max_{S \subseteq D}(P(S) - Q(S)) = \frac{1}{2} \sum_{x \in D} |P(x) - Q(x)| = \sum_{x \in D} \max(0, P(x) - Q(x)).$$

The TV distance satisfies many basic properties which makes it a versatile and fundamental measure for quantifying the dissimilarity between probability distributions. First, it has an explicit probabilistic interpretation: The TV distance between two distributions is the maximum gap between the probabilities assigned to a single event by the two distributions. Second, it satisfies many mathematically desirable properties: It is bounded and lies in $[0, 1]$, it is a metric, and it is invariant with respect to bijections. Third, it satisfies some interesting composability property. Given $f(g_1, g_2, \ldots, g_n)$, suppose we replace $g_2$ with $g_2'$ such that $d_{\mathrm{TV}}(g_2, g_2') \leq \varepsilon$. Then the TV distance between $f(g_1, g_2, \ldots, g_n)$ and $f(g_1, g_2', \ldots, g_n)$ is at most $\varepsilon$. Because of these reasons, the total variation distance is a central distance measure employed in a wide range of areas including probability and statistics Mitzenmacher & Upfal (2005), machine learning Shalev-Shwartz & Ben-David (2014), and information theory Cover & Thomas (2006), cryptography Stinson (1995), data privacy Dwork (2006), and pseudorandomness Vadhan (2012).

Lately, the computational aspects of TV distance have attracted a lot of attention. Sahai & Vadhan (2003) established, in a seminal work, that additively approximating the TV distance between two distributions that are samplable by Boolean circuits is hard for SZK (Statistical Zero Knowledge). The complexity class SZK is fundamental to cryptography and is believed to be computationally hard. Subsequent works captured variations of this theme Goldreich et al. (1999); Malka (2015); Dixon et al. (2020): For example, Goldreich et al. (1999) showed that the problem of deciding whether a distribution samplable by a Boolean circuit is close or far from the uniform distribution is complete for NISZK (Non-Interactive Statistical Zero Knowledge). Moreover, Cortes et al. (2007); Lyngsø & Pedersen (2002); Kiefer (2018) showed that it is undecidable to check whether the TV distance between two hidden Markov models is greater than a threshold or not, and that it is #P-hard to additively approximate it. Finally, Bhattacharyya et al. (2023) showed that $(a)$ exactly computing the

---

[*]Work done while the author was affiliated with the National University of Singapore.

TV distance between product distributions is #P-complete, and $(b)$ multiplicatively approximating the TV distance between Bayes nets is NP-hard.

On an algorithmic note, Bhattacharyya et al. (2020) designed efficient algorithms to additively approximate the TV distance between distributions efficiently samplable and efficiently computable (including the case of *ferromagnetic* Ising models). In particular, they designed efficient algorithms for additively approximating the TV distance of structured high dimensional distributions such as Bayesian networks, Ising models, and multivariate Gaussians. In a similar vein, Pote & Meel (2021) studied a related property testing variant of TV distance for distributions encoded by circuits. Multiplicative approximation of TV distance has received less attention compared to additive approximation. Recently, Bhattacharyya et al. (2023) gave an FPTAS for estimating the TV distance between an arbitrary product distribution and a product distribution with a bounded number of distinct marginals. Feng et al. (2023) designed an FPRAS for multiplicatively approximating the TV distance between two arbitrary product distributions and Feng et al. (2024) gave an FPTAS for the same task. Finally, Bhattacharyya et al. (2024) gave an FPRAS for estimating the TV distance between Bayes nets of small treewidth.

In this paper we address some previously unexplored (or under-explored) computational aspects of total variation distance relating to mixtures of product distributions and Ising models.

## 1.1 EQUIVALENCE CHECKING FOR MIXTURES OF PRODUCT DISTRIBUTIONS

Mixtures of product distributions constitute a natural and important class of distributions that have been studied in the mathematics and computer science literature. For instance, it is a standard observation that any distribution can be described by some (possibly large) mixture of product distributions (see Observation 9 in Appendix A).

Freund & Mansour (1999) gave an efficient algorithm for learning a mixture of two product distributions over the Boolean domain. As part of their analysis, they showed that given two mixtures of two product distributions, their KL divergence can be upper bounded by that of the components and a certain distance between the mixture coefficients. However, this upper bound does not lead to an equivalence checking algorithm.

A related problem in machine learning is source identification, whereby one is asked to identify the source parameters of a distribution. Gordon et al. (2021); Gordon & Schulman (2022); Gordon et al. (2023) give algorithms for source identification of a mixture of $k$ product distributions on $n$ bits, when given as input approximate values of multilinear moments.

We focus on the equivalence checking problem regarding mixtures of product distributions. Note that while it is easy to check whether two product distributions are equivalent, that is, by checking whether their respective Bernoulli parameters are equal, it is not clear how to do so for the case of mixtures of product distributions. This is so, because there are mixtures of product distributions that are equal (as distributions) but different sets of Bernoulli parameters describe them. For example, consider the case where we have two mixtures over one bit, namely $P = 1 \cdot P_1 + 0 \cdot P_2$ and $Q = \frac{1}{2} \cdot Q_1 + \frac{1}{2} \cdot Q_2$, where $P_1 = P_2 = \text{Bern}(\frac{1}{2})$ while $Q_1 = \text{Bern}(\frac{1}{3})$ and $Q_2 = \text{Bern}(\frac{2}{3})$. In this case, $P = Q = \text{Bern}(\frac{1}{2})$, but the parameters of $P$ and $Q$ are different.

We present a simple deterministic polynomial-time algorithm for checking equivalence between mixtures of product distributions. Let us first formally define mixtures of product distributions. Let $w_1, \ldots, w_k$ be real numbers (weights) such that $0 \le w_i \le 1$ for all $1 \le i \le k$, $\sum_{i=1}^{k} w_i = 1$, and $P_1, \ldots, P_k$ are $n$-dimensional product distributions over an alphabet $\Sigma$. The distribution $P$ specified by the tuple $(w_1, \ldots, w_k, P_1, \ldots, P_k)$ is a *mixture of products* if for all $x \in \Sigma^n$ it is the case that $P(x) = \sum_{i=1}^{k} w_i P_i(x)$. For a distribution $P$, we denote by $P^{\le i}$ its marginal on the first $i$ variables. We may now state our first main result.

**Theorem 1.** *There is a deterministic algorithm $E$ such that, given two mixtures of product distributions $P$ and $Q$, specified by $(w_1, \ldots, w_k, P_1, \ldots, P_k)$ and $(v_1, \ldots, v_k, Q_1, \ldots, Q_k)$, respectively, decides whether $P = Q$ or not. Moreover, if $P \neq Q$, then $E$ outputs some $x \in \Sigma^i$ (with $i \le n$) such that $P^{\le i}(x) \neq Q^{\le i}(x)$. The running time of $E$ is $O(nk^4|\Sigma|^4)$.*

(Note that the algorithm outlined in Theorem 1 has input size $\Omega(kn|\Sigma|)$.) The primary conceptual contribution of our work is a connection between equivalence checking for mixtures of distributions

and basis construction over appropriately chosen vector space. The connection lends itself to a construction that makes the algorithm as well as proof accessible to undergraduates.

## 1.2 Hardness of Approximating Total Variation Distance Between Ising Models

The Ising model (Ising, 1925; Lenz, 1920), originally developed to describe ferromagnetism in statistical mechanics, serves as a cornerstone in the study of phase transitions and critical phenomena. It consists of discrete variables, known as spins, which can take values of either $+1$ or $-1$. These spins are arranged on a lattice, and their interactions with nearest neighbors lead to a rich tapestry of behavior, including spontaneous magnetization and phase transitions at critical temperatures.

One of the most fascinating aspects of the Ising model is its ability to illustrate complex systems using simple rules. For instance, in 2D, it exhibits a second-order phase transition at a critical temperature, where the system changes from a disordered state to an ordered state as temperature decreases. This model has been extensively studied, leading to profound insights not only in physics but also in fields such as biology, sociology, and computer science. For more information the reader is invited to check the survey written by Cipra (1987).

On another note, the computational study of the Ising model has become increasingly relevant. With its relatively simple structure—interacting binary spins on a lattice, the Ising model serves as an ideal platform for exploring computational techniques ranging from Monte Carlo simulations to mean-field approximations. Monte Carlo methods, in particular, are widely used to investigate thermodynamic properties of the Ising model, as they allow for efficient sampling of spin configurations at various temperatures, enabling the computation of quantities like magnetization and susceptibility.

Some notable algorithmic results along these lines are the ones by Kasteleyn (1963) and Fisher (1966), whereby they showed that the evaluation of the partition function for planar Ising models can be reduced to some appropriate determinant computations. Moreover, Jerrum & Sinclair (1993) devised an efficient Monte Carlo approximation algorithm for estimating the partition function of arbitrary ferromagnetic (whereby all $w_{i,j}$'s are positive) Ising models.

On the other hand, there are some works pertaining to the intractability of computing various quantities of interest regarding Ising models (Welsh, 1993), such as the partition function outlined above. For example, Jerrum & Sinclair (1993) show that unless $\mathsf{NP} = \mathsf{RP}$, there is no fully polynomial-time randomized approximation scheme (FPRAS) to estimate the partition function of arbitrary Ising models. Moreover, Istrail (2000) proves that computing the partition function (for various kinds of Ising models) is NP-complete.

The second part of our work falls in this latter category. Let us first fix some notation. We focus on Ising models $P$ such that for all $x \in \{-1, 1\}^n$ it is the case that the probability that the underlying system of spins assumes the configuration $x$ is

$$P(x) = \frac{1}{Z} \exp\left(\sum_{i,j} w_{i,j} x_i x_j + \sum_i h_i x_i\right) \propto \exp\left(\sum_{i,j} w_{i,j} x_i x_j + \sum_i h_i x_i\right),$$

whereby $Z := \sum_y \exp\left(\sum_{i,j} w_{i,j} y_i y_j + \sum_i h_i y_i\right)$ is the partition function of $P$, and $\{w_{i,j}\}_{i,j}$ and $\{h_i\}_i$ are the parameters of the system.

We prove that it is hard to estimate the TV distance between Ising models under the very mild complexity-theoretic assumption $\mathsf{NP} \not\subseteq \mathsf{RP}$, which states that Boolean formula satisfiability (SAT) does not admit any one-sided-error randomized polynomial-time algorithm, that is, a randomized polynomial-time algorithm that may output a false positive answer with small probability (Arora & Barak, 2009).

**Theorem 2.** *If* $\mathsf{NP} \not\subseteq \mathsf{RP}$, *then there is no FPRAS that estimates the TV distance between any two Ising models.*

Our proof draws on the hardness result of Jerrum & Sinclair (1993), and shows that the partition function of Ising models can be reduced to the TV distance between Ising models, by a simple efficient *approximation preserving* reduction. The main ingredients of this reduction are as follows. First, we prove that estimating the partition function of any Ising model reduces to estimating any atomic

marginal the form $\mathbf{Pr}_P[x_k = \pm 1]$ for any variable $x_k$ and any Ising model $P$ (see Proposition 6). Then we show that estimating any atomic marginal the form $\mathbf{Pr}_P[x_k = \pm 1]$ for any variable $x_k$ and any Ising model $P$, can be reduced to estimating the TV distance between the Ising models $P, Q$, whereby $Q$ depends on $P$ (see Proposition 7).

## 1.3 PAPER ORGANIZATION

We give some preliminaries in Section 2. We prove Theorem 1 in Section 3 and Theorem 2 in Section 4. We conclude in Section 5 with some interesting open problems. Observation 9 is proved in Appendix A and Proposition 6 is proved in Appendix B.

## 2 PRELIMINARIES

We require the following folklore result, which is an application of Gaussian elimination.

**Proposition 3.** *There is a deterministic algorithm $G$ that gets as input a set of vectors $V$, and outputs a maximum-size subset $S \subseteq V$ of linearly independent vectors. The running time of $G$ is $O\left(|V|^4\right)$.*

An $n$-dimensional product distribution $R$ over an alphabet $\Sigma$ is described by the $n \, |\Sigma|$ parameters

$$\left\{ \mathbf{Pr}_R[X_i = y] \right\}_{\substack{i \in [n], \\ y \in \Sigma}}, \qquad \text{so that} \qquad R(x) = \prod_{i=1}^{n} \mathbf{Pr}_R[X_i = x_i] \qquad \text{for all } x \in \Sigma^n.$$

For $n$-dimensional product distribution $R$ over an alphabet $\Sigma$, we denote its marginal over the first $1 \leq j \leq n$ coordinates by $R^{\leq j}$. Note that for any $x \in \Sigma^j$ we have $R^{\leq j}(x) = \prod_{i=1}^{j} \mathbf{Pr}_R[X_i = x_i]$.

We shall also require the following notion of approximation algorithm.

**Definition 4.** A function $f : \{0, 1\}^* \to \mathbb{R}$ admits a *fully polynomial-time randomized approximation scheme (FPRAS)* if there is a *randomized* algorithm $\mathcal{A}$ such that for every input $x$ (of length $n$) and parameters $\varepsilon, \delta > 0$, the algorithm $\mathcal{A}$ outputs a $\varepsilon$-multiplicative approximation of $f(x)$, i.e., a value that lies in the interval $[f(x)/(1 + \varepsilon), (1 + \varepsilon)f(x)]$, with probability at least $1 - \delta$. The running time of $\mathcal{A}$ is polynomial in $n, 1/\varepsilon, 1/\delta$.

## 3 EQUIVALENCE CHECKING FOR MIXTURES OF PRODUCT DISTRIBUTIONS

Let us now prove Theorem 1. First, observe that $P = Q$ if and only if $P^{\leq j} = Q^{\leq j}$ for all $1 \leq j \leq n$. This is so, since if $P = Q$, then every marginal of $P$ matches the respective marginal of $Q$ (in symbols, $P^{\leq j} = Q^{\leq j}$ for all $1 \leq j \leq n$). Otherwise, there would be some $1 \leq j \leq n$ and $y \in \Sigma^j$ such that $P^{\leq j}(y) \neq Q^{\leq j}(y)$. The latter would then establish the existence of an $x := (y, z) \in \Sigma^n$ (for some $z \in \Sigma^{n-j}$) such that $P(x) \neq Q(x)$. On the other hand, if $P^{\leq j} = Q^{\leq j}$ for all $1 \leq j \leq n$, then $P^{\leq n} = Q^{\leq n}$, which in particular implies that $P = Q$.

Note that the condition $P^{\leq j} = Q^{\leq j}$ for all $1 \leq j \leq n$ is equivalent, by the definitions of $P$ and $Q$, to the condition $\sum_{i=1}^{k} w_i P_i^{\leq j} = \sum_{i=1}^{k} v_i Q_i^{\leq j}$ for all $1 \leq j \leq n$. Thus, if $P \neq Q$, then there is some $1 \leq j \leq n$ so that $\sum_{i=1}^{k} w_i P_i^{\leq j} \neq \sum_{i=1}^{k} v_i Q_i^{\leq j}$.

We will use an inductive argument on $1 \leq j \leq n$ to show that these conditions can be efficiently checked (in either case).

**Base Case.** For $j = 1$, we can efficiently check whether it is the case that

$$\sum_{i=1}^{k} w_i P_i^{\leq 1} = \sum_{i=1}^{k} v_i Q_i^{\leq 1}.$$

This is done by checking for all $x \in \Sigma$ that

$$\sum_{i=1}^{k} w_i \mathbf{Pr}_{P_i}[X_1 = x] - \sum_{i=1}^{k} v_i \mathbf{Pr}_{Q_i}[X_1 = x] = 0.$$

If these tests pass, then the algorithm proceeds with the inductive argument outlined below (otherwise, it outputs $x$). Towards this, we will now find a basis $B_1$ for the set of coefficient vectors of the equations

$$\left\{ \sum_{i=1}^{k} w_i \Pr_{P_i}[X_1 = x] \, z_i - \sum_{i=1}^{k} v_i \Pr_{Q_i}[X_1 = x] \, z_{k+i} = 0 \right\}_{x \in \Sigma},$$

over variables $z_1, \dots, z_{2k}$. Note that the size of $B_1$ is at most $\min(2k, |\Sigma|) \leq 2k$. We can find $B_1$ as follows. We appeal to Proposition 3 and run the algorithm $G$ outlined there on the set of vectors

$$\left\{ \left( w_1 \Pr_{P_1}[X_1 = x], \dots, w_k \Pr_{P_k}[X_1 = x], -v_1 \Pr_{Q_1}[X_1 = x], \dots, -v_k \Pr_{Q_k}[X_1 = x] \right) \right\}_{x \in \Sigma}$$

(in time $O\left(|\Sigma|^4\right)$). Then, we define $B_1$ to be the set of independent vectors being output by $G$.

**Induction Hypothesis.**  Assume that for a $j \geq 1$ it is the case that

$$\sum_{i=1}^{k} w_i P_i^{\leq j} = \sum_{i=1}^{k} v_i Q_i^{\leq j},$$

and we have a basis $B_j$ for the set of coefficient vectors of the following equations

$$\left\{ \sum_{i=1}^{k} w_i P_i^{\leq j}(x) \, z_i - \sum_{i=1}^{k} v_i Q_i^{\leq j}(x) \, z_{k+i} = 0 \right\}_{x \in \Sigma^j}$$

over variables $z_1, \dots, z_{2k}$. Note that $B_j$ is of size at most $\min\left(2k, |\Sigma|^j\right) \leq 2k$.

**Induction Step.**  We will establish that we can check whether $P$ and $Q$ agree up to coordinate $j+1$ and compute a basis $B_{j+1}$ for the respective set of coefficient vectors of the equations that capture this equivalence.

To see whether $P$ and $Q$ agree up to coordinate $j+1$, one needs to check that

$$\sum_{i=1}^{k} w_i P_i^{\leq j}(x) \Pr_{P_i}[X_{j+1} = y] - \sum_{i=1}^{k} v_i Q_i^{\leq j}(x) \Pr_{Q_i}[X_{j+1} = y] = 0$$

for all $x \in \Sigma^j$ and $y \in \Sigma$. A crucial observation (that follows from the inductive hypothesis) is that we only need to check whether these equations hold for the assignments $x$ that correspond to vectors in $B_j$, and the values $y \in \Sigma$. (Note that each basis vector $b \in B_j$ can be specified by an assignment $x_b \in \Sigma^j$. This follows from the way these basis vectors are constructed. See below, the discussion after Claim 5.) If any of these tests fails, then the algorithm outputs $(x, y)$; else, it continues as follows.

To proceed with the induction, it would suffice to show how to construct a basis $B_{j+1}$ for the set of coefficient vectors of the following equations over variables $z_1, \dots, z_{2k}$, namely

$$\left\{ \sum_{i=1}^{k} w_i P_i^{\leq j}(x) \Pr_{P_i}[X_{j+1} = y] \, z_i - \sum_{i=1}^{k} v_i Q_i^{\leq j}(x) \Pr_{Q_i}[X_{j+1} = y] \, z_{k+i} = 0 \right\}_{\substack{x \in \Sigma^j, \\ y \in \Sigma}}.$$

Let $B_j = \{ b_1 = (b_{1,1}, \dots, b_{1,2k}), \dots, b_m = (b_{m,1}, \dots, b_{m,2k}) \}$ whereby $m \leq 2k$ and $C := \bigcup_{i=1}^{m} C_i$ is such that

$$C_1 := \left\{ \left( b_{1,1} \Pr_{P_1}[X_{j+1} = y], \dots, b_{1,k} \Pr_{P_k}[X_{j+1} = y], \right.\right.$$

$$\left.\left. b_{1,k+1} \Pr_{Q_1}[X_{j+1} = y], \dots, b_{1,2k} \Pr_{Q_k}[X_{j+1} = y] \right) \right\}_{y \in \Sigma},$$

$$\vdots$$

$$C_m := \left\{ \left( b_{m,1} \Pr_{P_1}[X_{j+1} = y], \ldots, b_{m,k} \Pr_{P_k}[X_{j+1} = y], \right.\right.$$
$$\left.\left. b_{m,k+1} \Pr_{Q_1}[X_{j+1} = y], \ldots, b_{m,2k} \Pr_{Q_k}[X_{j+1} = y] \right) \right\}_{y \in \Sigma}.$$

We require the following claim. Below, we denote the dot product of vectors $a, b \in \mathbb{R}^s$ by $\langle a, b \rangle$. That is, $\langle a, b \rangle = \sum_{i=1}^s a_i b_i$.

**Claim 5.** *It is the case that $B_{j+1} \subseteq C$.*

Claim 5 helps explain how, for any vector $b \in B_{j+1}$, one may extract an assignment $x_b \in \Sigma^{j+1}$ that corresponds to $b$. This is done by keeping track of the Bernoulli parameters that appear in $b$: There is exactly one such parameter for each variable, and it refers to a symbol in $\Sigma$.

*Proof of Claim 5.* Let $y \in \Sigma$. We have that

$$\sum_{i=1}^k w_i P_i^{\leq j+1}(x, y) z_i - \sum_{i=1}^k v_i Q_i^{\leq j+1}(x, y) z_{k+i}$$
$$= \sum_{i=1}^k w_i P_i^{\leq j}(x) \Pr_{P_i}[X_{j+1} = y] z_i - \sum_{i=1}^k v_i Q_i^{\leq j}(x) \Pr_{Q_i}[X_{j+1} = y] z_{k+i}.$$

Let $z := (z_1, \ldots, z_{2k})$. By the inductive hypothesis, for any $x \in \Sigma^j$,

$$\sum_{i=1}^k w_i P_i^{\leq j}(x) z_i - \sum_{i=1}^k v_i Q_i^{\leq j}(x) z_{k+i}$$

can be written as a linear combination of $\{\langle b_\ell, z \rangle\}_{\ell=1}^m$. Concretely, there exist $d_1, \ldots, d_m \in \mathbb{R}$ such that

$$\sum_{i=1}^k w_i P_i^{\leq j}(x) z_i - \sum_{i=1}^k v_i Q_i^{\leq j}(x) z_{k+i} = \sum_{\ell=1}^m d_\ell \langle b_\ell, z \rangle.$$

Since the corresponding coefficients of each $z_i$ must be equal between the LHS and the RHS of this equation, we get that for all $1 \leq i \leq k$,

$$w_i P_i^{\leq j}(x) = \sum_{\ell=1}^m d_\ell \cdot b_{\ell,i} \qquad \text{and} \qquad v_i Q_i^{\leq j}(x) = -\sum_{\ell=1}^m d_\ell \cdot b_{\ell,k+i} \tag{1}$$

for some $d_1, \ldots, d_m$. Then it is straightforward to see that for every $y \in \Sigma$

$$\sum_{i=1}^k w_i P_i^{\leq j}(x) \Pr_{P_i}[X_{j+1} = y] z_i - \sum_{i=1}^k v_i Q_i^{\leq j}(x) \Pr_{Q_i}[X_{j+1} = y] z_{k+i} = \sum_{\ell=1}^m d_\ell \langle c_\ell, z \rangle,$$

whereby $c_\ell$ is the vector within the set $C_\ell$ that corresponds to the setting where $X_{j+1} = y$. To see why this equation holds, we will consider the coefficients of each $z_i$ in either side of this equation and show that they are equal. Let us consider the case where $1 \leq i \leq k$. (The case where $k + 1 \leq i \leq 2k$ is similar.) In the LHS, the coefficient of $z_i$ is equal to

$$w_i P_i^{\leq j}(x) \Pr_{P_i}[X_{j+1} = y] = \sum_{\ell=1}^m d_\ell \cdot b_{\ell,i} \cdot \Pr_{P_i}[X_{j+1} = y],$$

by Equation (1), while in the RHS it is equal to

$$\sum_{\ell=1}^m d_\ell \cdot c_{\ell,i} = \sum_{\ell=1}^m d_\ell \cdot b_{\ell,i} \cdot \Pr_{P_i}[X_{j+1} = y],$$

by the definition of $c_\ell$. This shows that $B_{j+1}$ is a subset of $C$, and the proof is complete. $\square$

By Claim 5, we have that $B_{j+1}$ contains (at most) $m |\Sigma| = O(k |\Sigma|)$ vectors. However, not all of these $O(k |\Sigma|)$ vectors are (necessarily) independent. By Proposition 3, we can find the (at most) $2k$ independent vectors among the $O(k |\Sigma|)$ vectors in time $O\left(k^4 |\Sigma|^4\right)$. These vectors constitute $B_{j+1}$. This concludes the inductive description of our algorithm.

**Running Time.** Let $T(n, k, \Sigma)$ denote the running time of this procedure, on mixtures of size $k$ over $n$ variables that have alphabet $\Sigma$. We have the recurrence relation

$$T(n, k, \Sigma) \leq T(n-1, k, \Sigma) + O\left(k^4 |\Sigma|^4\right),$$

which in particular yields $T(n, k, \Sigma) = O\left(nk^4 |\Sigma|^4\right)$.

# 4 HARDNESS OF APPROXIMATING TOTAL VARIATION DISTANCE BETWEEN ISING MODELS

In this section we prove Theorem 2.

## 4.1 REDUCING THE PARTITION FUNCTION TO ATOMIC MARGINALS

We first observe the following. The proof is standard and is given in Appendix B.

**Proposition 6.** *If there is some FPRAS that estimates any atomic marginal of any Ising model $P$, namely $\mathbf{Pr}_P[x_k = \pm 1]$ for any variable $x_k$, then there is some FPRAS that estimates the partition function of any Ising model.*

## 4.2 REDUCING THE ATOMIC MARGINALS TO TV DISTANCE

We prove the following.

**Proposition 7.** *If there is some FPRAS that estimates the TV distance between any two Ising models, then there is some FPRAS that estimates any atomic marginal of any Ising model $P$, namely $\mathbf{Pr}_P[x_k = \pm 1]$ for any variable $x_k$.*

*Proof.* Assume that there is some FPRAS that estimates the TV distance between any two Ising models. We will show that there is some FPRAS that estimates the atomic marginals of any Ising model $P$, namely $\mathbf{Pr}_P[x_k = \pm 1]$ for any variable $x_k$. Let $\varepsilon$ be the desired accuracy error of the FPTAS that estimates the atomic marginals of any Ising model $P$.

Fix some Ising model $P$ with parameters $\{w_{i,j}\}_{i,j}$ and $\{h_i\}_i$. We shall first introduce a new dummy variable $x_0$. Let $P_0$ be a new Ising model over $x_0, \ldots, x_n$ with parameters $\{w_{i,j}\}_{i,j}$ and $\{h_i\}_i$ so that $w_{0,i} = 0$ for all $i > 0$ and $h_0 \to -\infty$ (that is, $h_0$ is a small negative quantity which we will fix later). Note that under these conditions $x_0$ is independent from every other node $x_1, \ldots, x_n$. Moreover,

$$
\begin{aligned}
\mathbf{Pr}_{P_0}[x_0 = 1] = \sum_{x:x_0=1} P_0(x) &= \frac{\sum_{x:x_0=1} \exp\left(\sum_{i,j} w_{i,j} x_i x_j + \sum_i h_i x_i\right)}{\sum_x \exp\left(\sum_{i,j} w_{i,j} x_i x_j + \sum_i h_i x_i\right)} \\
&= \frac{\sum_{x:x_0=1} \exp\left(\sum_{i,j} w_{i,j} x_i x_j + h_0 x_0 + \sum_{i>0} h_i x_i\right)}{\sum_x \exp\left(\sum_{i,j} w_{i,j} x_i x_j + h_0 x_0 + \sum_{i>0} h_i x_i\right)} \\
&= \frac{\sum_{x:x_0=1} \exp\left(\sum_{i,j} w_{i,j} x_i x_j + h_0 + \sum_{i>0} h_i x_i\right)}{\sum_x \exp\left(\sum_{i,j} w_{i,j} x_i x_j + h_0 x_0 + \sum_{i>0} h_i x_i\right)} \\
&= \frac{\exp(h_0) \sum_{x:x_0=1} \exp\left(\sum_{i,j} w_{i,j} x_i x_j + \sum_{i>0} h_i x_i\right)}{\sum_x \exp\left(\sum_{i,j} w_{i,j} x_i x_j + h_0 x_0 + \sum_{i>0} h_i x_i\right)} \\
&= \frac{\exp(h_0) \sum_x \exp\left(\sum_{i,j} w_{i,j} x_i x_j + \sum_{i>0} h_i x_i\right)}{\sum_x \exp\left(\sum_{i,j} w_{i,j} x_i x_j + h_0 x_0 + \sum_{i>0} h_i x_i\right)}.
\end{aligned}
$$

Let now us define $E(x) := \exp\left(\sum_{i,j} w_{i,j} x_i x_j + \sum_{i>0} h_i x_i\right)$ for all $x$. We have

$$\Pr_{P_0}[x_0 = 1] = \frac{\exp(h_0)\sum_x E(x)}{\exp(h_0)\sum_{x:x_0=1} E(x) + \exp(-h_0)\sum_{x:x_0=-1} E(x)}$$

$$= \frac{\exp(h_0)\sum_x E(x)}{\exp(h_0)\sum_x E(x) + \exp(-h_0)\sum_x E(x)}$$

$$= \frac{\exp(2h_0)\sum_x E(x)}{\exp(2h_0)\sum_x E(x) + \sum_x E(x)} = \frac{\exp(2h_0)}{\exp(2h_0)+1}.$$

We shall now define another Ising model $Q$ over $x_0, \ldots, x_n$ as follows. The model $Q$ has parameters $\{w'_{i,j}\}_{i,j}$ and $\{h'_i\}_i$ so that $w'_{i,j} := w_{i,j}$ if $(i,j) \neq (0,k)$ and $w'_{0,k} := w_{0,k} + \delta$ for some $\delta > 1$, and $h'_i = h_i$ for all $i$. We have that for all $x \in \{-1,1\}^n$ it is the case that

$$Q(x) \propto \exp\left(\sum_{i,j} w'_{i,j} x_i x_j + \sum_i h'_i x_i\right)$$

$$= \exp\left(\sum_{i,j} w'_{i,j} x_i x_j + \sum_i h_i x_i\right)$$

$$= \exp\left((w_{0,k} + \delta) x_0 x_k + \sum_{(i,j)\neq(0,k)} w_{i,j} x_i x_j + \sum_i h_i x_i\right)$$

$$= \exp\left(w_{0,k} x_0 x_k + \delta x_0 x_k + \sum_{(i,j)\neq(0,k)} w_{i,j} x_i x_j + \sum_i h_i x_i\right)$$

$$= \exp\left(\delta x_0 x_k + \sum_{i,j} w_{i,j} x_i x_j + \sum_i h_i x_i\right)$$

$$= \exp(\delta x_0 x_k)\exp\left(\sum_{i,j} w_{i,j} x_i x_j + \sum_i h_i x_i\right) = \exp(\delta x_0 x_k)\, P_0(x)\, Z_{P_0},$$

whereby $Z_{P_0}$ is the partition function of $P_0$. If $x_0 x_k = 1$, then $Q(x) \propto \exp(\delta)\, P_0(x)$; else $Q(x) \propto \exp(-\delta)\, P_0(x)$. Moreover, for any $x$ such that $x_0 x_k = 1$,

$$Q(x) = \frac{\exp(\delta)\, P_0(x)\, Z_{P_0}}{Z_Q}$$

$$= \frac{\exp(\delta)\, P_0(x)\, Z_{P_0}}{\sum_{x:x_0 x_k=1} \exp(\delta)\, P_0(x)\, Z_{P_0} + \sum_{x:x_0 x_k=-1} \exp(-\delta)\, P_0(x)\, Z_{P_0}}$$

$$= \frac{P_0(x)}{\sum_{x:x_0 x_k=1} P_0(x) + \exp(-2\delta)\sum_{x:x_0 x_k=-1} P_0(x)}$$

$$\geq \frac{P_0(x)}{\sum_{x:x_0 x_k=1} P_0(x) + \sum_{x:x_0 x_k=-1} P_0(x)} = \frac{P_0(x)}{\sum_x P_0(x)} \geq P_0(x)$$

where $Z_Q$ is the partition function of $Q$. Similarly, for any $x$ such that $x_0 x_k = -1$,

$$Q(x) = \frac{\exp(-\delta)\, P_0(x)\, Z_{P_0}}{Z_Q}$$

$$= \frac{\exp(-\delta)\, P_0(x)\, Z_{P_0}}{\sum_{x:x_0 x_k=1} \exp(\delta)\, P_0(x)\, Z_{P_0} + \sum_{x:x_0 x_k=-1} \exp(-\delta)\, P_0(x)\, Z_{P_0}}$$

$$= \frac{P_0(x)}{\sum_{x:x_0 x_k=1} \exp(2\delta)\, P_0(x) + \sum_{x:x_0 x_k=-1} P_0(x)}$$

$$< \frac{P_0(x)}{\sum_{x:x_0 x_k = 1} P_0(x) + \sum_{x:x_0 x_k = -1} P_0(x)} = P_0(x).$$

That is, $P_0(x) \geq Q(x)$ if and only if $x_0 x_k = -1$. We now have

$$d_{\mathrm{TV}}(P_0, Q) = \sum_x \max(0, P_0(x) - Q(x))$$

$$= \sum_{x:P_0(x) \geq Q(x)} (P_0(x) - Q(x))$$

$$= \sum_{x:x_0 x_k = -1} (P_0(x) - Q(x)) = \sum_{x:x_0 x_k = -1} P_0(x) - \sum_{x:x_0 x_k = -1} Q(x).$$

Moreover,

$$\sum_{x:x_0 x_k = -1} P_0(x) = \Pr_{P_0}[x_0 x_k = -1]$$

$$= \Pr_{P_0}[x_0 = 1, x_k = -1] + \Pr_{P_0}[x_0 = -1, x_k = 1]$$

$$= \Pr_{P_0}[x_0 = 1] \Pr_{P_0}[x_k = -1] + \Pr_{P_0}[x_0 = -1] \Pr_{P_0}[x_k = 1]$$

$$= \Pr_{P_0}[x_0 = 1] \Pr_{P}[x_k = -1] + \Pr_{P_0}[x_0 = -1] \Pr_{P}[x_k = 1]$$

$$= \Pr_{P_0}[x_0 = 1] + \left(1 - 2\Pr_{P_0}[x_0 = 1]\right) \Pr_{P}[x_k = 1]$$

$$= \frac{\exp(2h_0)}{\exp(2h_0) + 1} + \frac{1 - \exp(2h_0)}{\exp(2h_0) + 1} \Pr_{P}[x_k = 1],$$

whereby we have used the fact that $\mathbf{Pr}_{P_0}[x_k = \pm 1] = \mathbf{Pr}_P[x_k = \pm 1]$. That is,

$$d_{\mathrm{TV}}(P_0, Q) = \frac{\exp(2h_0)}{\exp(2h_0) + 1} - \sum_{x:x_0 x_k = -1} Q(x) + \frac{1 - \exp(2h_0)}{\exp(2h_0) + 1} \Pr_{P}[x_k = 1]. \tag{2}$$

Note that

$$0 \leq \sum_{x:x_0 x_k = -1} Q(x) = \sum_{x:x_0 x_k = -1} \frac{\exp(-\delta) P_0(x) Z_{P_0}}{Z_Q} \leq \frac{\exp(-\delta) Z_{P_0}}{Z_Q}.$$

Intuitively, note that if (in the above) we set $\delta \to \infty$ and $h_0 \to -\infty$, then $d_{\mathrm{TV}}(P_0, Q) = \mathbf{Pr}_P[x_k = 1]$, and an approximation of $d_{\mathrm{TV}}(P_0, Q)$ implies an approximation of $\mathbf{Pr}_P[x_k = 1]$. Now we make this formal by finitely quantifying $h_0, \delta$.

It would suffice to show that

$$\left| d_{\mathrm{TV}}(P_0, Q) - \Pr_{P}[x_k = 1] \right| \leq \frac{\varepsilon}{2} \cdot \Pr_{P}[x_k = 1].$$

Then this could be combined with an $\frac{\varepsilon}{2}$-multiplicative approximation of $d_{\mathrm{TV}}(P_0, Q)$ to yield the desired $\varepsilon$-multiplicative approximation of $\mathbf{Pr}_P[x_k = 1]$. By Equation (2),

$$\left| d_{\mathrm{TV}}(P_0, Q) - \Pr_{P}[x_k = 1] \right| \leq \frac{\exp(2h_0)}{\exp(2h_0) + 1} + \frac{\exp(-\delta) Z_{P_0}}{Z_Q} + \frac{\exp(2h_0)}{\exp(2h_0) + 1} \Pr_{P}[x_k = 1]$$

$$\leq 2\exp(2h_0) + \frac{\exp(-\delta) Z_{P_0}}{Z_Q}.$$

Let $W := \max_{i,j} |w_{i,j}|$ and $H := \max_i |h_i|$. Then for any $x$,

$$\Pr_{P}[x_k = 1] = \sum_{x:x_k = 1} \frac{\exp\left(\sum_{i,j} w_{i,j} x_i x_j + \sum_i h_i x_i\right)}{\sum_y \exp\left(\sum_{i,j} w_{i,j} y_i y_j + \sum_i h_i y_i\right)}$$

$$\geq \frac{2^{n-1} \exp\left(-W(n+1)^2 - (n+1)H\right)}{2^{n+1} \exp\left(W(n+1)^2 + (n+1)H\right)}$$

$$= \frac{\exp\left(-W\left(n+1\right)^2 - \left(n+1\right)H\right)}{4\exp\left(W\left(n+1\right)^2 + \left(n+1\right)H\right)}.$$

Choosing $h_0$ and $\delta$ such that $\max(-h_0, \delta) = \Omega(\text{poly}(n, H, W, 1/\varepsilon, Z_{P_0}/Z_Q)) = \Omega(\text{poly}(n, H, W, 1/\varepsilon))$ (note that $Z_{P_0}/Z_Q$ can be bounded by some polynomial in $n, H, W$) we may ensure the desired

$$\frac{\exp\left(-W\left(n+1\right)^2 - \left(n+1\right)H\right)}{4\exp\left(W\left(n+1\right)^2 + \left(n+1\right)H\right)} \geq \frac{2}{\varepsilon}\left(2\exp(2h_0) + \frac{\exp\left(-\delta\right)Z_{P_0}}{Z_Q}\right).$$

An identical to the above argument can be employed to get a multiplicative approximation of $\mathbf{Pr}_P[x_k = -1]$ (by letting $h_0 \to \infty$ above).

Since $d_{\text{TV}}(P_0, Q)$ multiplicatively approximates $\mathbf{Pr}_P[x_k = \pm 1]$, this is an approximation preserving reduction, as any multiplicative approximation of $d_{\text{TV}}(P_0, Q)$ is a multiplicative approximation of $\mathbf{Pr}_P[x_k = \pm 1]$. $\qquad\square$

### 4.3 PROOF OF THEOREM 2

To prove Theorem 2 we shall require the following result of Jerrum & Sinclair (1993).

**Theorem 8** (Jerrum & Sinclair (1993))**.** *If* NP $\not\subseteq$ RP*, then there is no FPRAS that estimates the partition function of any Ising model.*

We now turn to the proof of Theorem 2.

*Proof of Theorem 2.* By Proposition 6, if there exists some FPRAS that estimates any atomic marginal of any Ising model, then there exists some FPRAS that estimates the partition function of any Ising model. By Proposition 7, if there exists some FPRAS that estimates the TV distance between Ising models, then there exists some FPRAS that estimates any atomic marginal of any Ising model. That is, if there exists some FPRAS that estimates the TV distance between Ising models, then there exists some FPRAS that estimates the partition function of any Ising model. By Theorem 8, if NP $\not\subseteq$ RP, then there is no FPRAS that estimates the partition function of any Ising model. That is, if NP $\not\subseteq$ RP, then there is no FPRAS that estimates the TV distance between Ising models. This concludes the proof. $\qquad\square$

## 5 DISCUSSION

We have shown how to efficiently check whether two mixtures of product distributions are equivalent or not. One might wonder whether our ideas can be applied to check equivalence between mixtures of other, more expressive, kinds of distributions, such as Bayes nets.

Bayes nets constitute Pearl (1989) a probabilistic graphical model, with numerous applications in machine learning and inference, that naturally generalize product distributions in that they allow for dependencies among different variables. Bhattacharyya et al. (2023) showed that it is NP-hard to decide whether the total variation distance between two Bayes nets $P$ and $Q$ is equal to $0$ or not, so one cannot hope to extend our methods to testing equivalence between mixtures of Bayes net distributions (unless, of course, P = NP.) Can our ideas be extended to testing equivalence between mixtures of some subclass of Bayes net distributions, such as Bayes net distributions whereby their underlying graph is a tree or has small treewidth?

Our second result, namely the hardness of approximating the TV distance between Ising models, helps us further understand the intricacies of the Ising model and the consequences of complexity-theoretic conjectures such as NP $\not\subseteq$ RP. Can we extend this complexity-theoretic hardness of approximation to other classes of probability distributions, such as factor graphs or general undirected probabilistic graphical models?

ACKNOWLEDGEMENTS

DesCartes: This research is supported by the National Research Foundation, Prime Minister's Office, Singapore under its Campus for Research Excellence and Technological Enterprise (CREATE) programme. This work was supported in part by National Research Foundation Singapore under its NRF Fellowship Programme [NRF-NRFFAI1-2019-0004] and an Amazon Research Award. We acknowledge the support of the Natural Sciences and Engineering Research Council of Canada (NSERC), [funding reference number RGPIN-2024-05956]. The work of AB was supported in part by National Research Foundation Singapore under its NRF Fellowship Programme (NRF-NRFFAI-2019-0002) and an Amazon Faculty Research Award. The work of SG was supported in part by the SERB award CRG/2022/007985. Pavan's work is partly supported by NSF awards 2130536, 2342245, 2413849. Vinod's work is partly supported by NSF awards 2130608, 2342244, 2413848.

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

## A  THE EXPRESSIBILITY OF MIXTURES OF PRODUCTS

We require the following observation.

**Observation 9.** *Let $D$ be a distribution over $\Sigma^n$ for some alphabet $\Sigma$. Then for $k := |\Sigma^n|$ there exists a mixture of product distributions $(w_1, \ldots, w_k, P_1, \ldots, P_k)$ over $\Sigma^n$ such that for all $x \in \Sigma^n$ it is the case that*

$$D(x) = \sum_{i=1}^{k} w_i P_i(x) \,.$$

*Proof.* Let $x_1, \ldots, x_k$ be the elements of $\Sigma^n$. For all $1 \le i \le k$, let $P_i$ be a product distribution on $n$ variables such that $P_i(y) = 1$ if $y = x_i$, otherwise, $P_i(y) = 0$. We can construct such a product distribution $P_i$ as follows. First, note that $P_i(y) = \prod_{j=1}^{n} p_{i,j}(y_j)$, whereby $p_{i,j}(z)$ is the probability that the $j$-th coordinate of $P_i$ takes the value $z \in \Sigma$. Let $x_i = x_{i,1}, \ldots, x_{i,n}$, whereby $x_{i,j} \in \Sigma$. Then for all $j$ define $p_{i,j}(z) := 1$ if $z = x_{i,j}$, otherwise, $p_{i,j}(z) := 0$. Moreover, let $w_i := D(x_i)$ and note that $\sum_{i=1}^{k} w_i = 1$. Then the desired equality follows by straightforward calculations. $\square$

## B  PROOF OF PROPOSITION 6

Let us first assume that there is some FPRAS $M$ that runs in time $t_M$ and estimates any atomic marginal of any Ising model. We will prove there is some FPRAS that estimates the partition function of any Ising model.

Let $P_1$ be an Ising model over variables $x_1, \ldots, x_n$ with parameters $\{w_{i,j}\}_{i,j}$ and $\{h_i\}_i$ with partition function $Z_1$. We will show how to estimate $Z_1$. To this end, let us define a new Ising model $P_2$ over variables $x_2, \ldots, x_n$ with parameters $\left\{ w_{i,j}^{(2)} \right\}_{i,j}$ and $\left\{ h_i^{(2)} \right\}_i$ so that $w_{i,j}^{(2)} := w_{i,j}$ and $h_i^{(2)} := w_{1,i} + h_i$ for all $2 \le i < j \le n$. Let $Z_2$ be the partition function of $P_2$. Let also

$$E_1(x) := \exp\left( \sum_{i,j} w_{i,j} x_i x_j + \sum_i h_i x_i \right), \qquad E_2(x) := \exp\left( \sum_{i,j \ne 1} w_{i,j}^{(2)} x_i x_j + \sum_{i \ne 1} h_i^{(2)} x_i \right),$$

for all $x$, and note that $Z_1 = \sum_x E_1(x)$ and $Z_2 = \sum_x E_2(x)$. We have

$$\sum_{x:x_1=1} E_1(x) = \sum_{x:x_1=1} \exp\left( \sum_{i,j} w_{i,j} x_i x_j + \sum_i h_i x_i \right)$$

$$= \sum_{x:x_1=1} \exp\left( \sum_{i,j \ne 1} w_{i,j} x_i x_j + \sum_i w_{1,i} x_i + \sum_{i \ne 1} h_i x_i + h_1 \right)$$

$$= \sum_{x:x_1=1} \exp\left( \sum_{i,j \ne 1} w_{i,j} x_i x_j + \sum_{i \ne 1} (w_{1,i} + h_i) x_i + h_1 \right)$$

$$= \sum_{x:x_1=1} \exp\left( \sum_{i,j \ne 1} w_{i,j} x_i x_j + \sum_{i \ne 1} (w_{1,i} + h_i) x_i \right) \exp(h_1)$$

$$= \exp(h_1) \sum_{x:x_1=1} \exp\left( \sum_{i,j \ne 1} w_{i,j} x_i x_j + \sum_{i \ne 1} (w_{1,i} + h_i) x_i \right)$$

$$= \exp(h_1) \sum_{x:x_1=1} \exp\left( \sum_{i,j \ne 1} w_{i,j}^{(2)} x_i x_j + \sum_{i \ne 1} h_i^{(2)} x_i \right)$$

$$= \exp(h_1) \sum_{x} \exp\left( \sum_{i,j \ne 1} w_{i,j}^{(2)} x_i x_j + \sum_{i \ne 1} h_i^{(2)} x_i \right)$$

$$= \exp(h_1) \sum_x E_2(x) = \exp(h_1) \, Z_2.$$

Moreover,

$$\mathbf{Pr}_{P_1}[x_1 = 1] = \frac{1}{Z_1} \sum_{x:x_1=1} E_1(x) \qquad \text{or} \qquad Z_1 = \frac{1}{\mathbf{Pr}_{P_1}[x_1 = 1]} \sum_{x:x_1=1} E_1(x) \, .$$

Combining the above, we have that

$$Z_1 = \frac{1}{\mathbf{Pr}_{P_1}[x_1 = 1]} \sum_{x:x_1=1} E_1(x) = Z_2 \frac{\exp(h_1)}{\mathbf{Pr}_{P_1}[x_1 = 1]}.$$

This equality yields a natural recursive algorithm for computing $Z_1$, whereby the computation of $Z_1$ is reduced to the computation of $Z_2$ by making use of the algorithm $M$ that estimates the marginal $\mathbf{Pr}_{P_1}[x_1 = 1]$. This continues until we reach in the recursion some Ising model $P_n$ over the variable $x_n$. At this point the computation of $Z_n$ is trivial, as

$$Z_n = \exp\left(h_n^{(n)}\right) + \exp\left(-h_n^{(n)}\right).$$

We will now bound the approximation and confidence errors of this algorithm. Note that

$$Z_1 = Z_2 \frac{\exp(h_1)}{\mathbf{Pr}_{P_1}[x_1 = 1]} = \cdots = Z_n \frac{\exp(h_1) \cdot \exp\left(h_2^{(2)}\right) \cdot \ldots \cdot \exp\left(h_{n-1}^{(n-1)}\right)}{\mathbf{Pr}_{P_1}[x_1 = 1] \cdot \mathbf{Pr}_{P_2}[x_2 = 1] \cdot \ldots \cdot \mathbf{Pr}_{P_{n-1}}[x_{n-1} = 1]}$$

whereby each atomic marginal in the denominator is approximated with a $(1 + \varepsilon_0)$ ratio for $\varepsilon_0 := \varepsilon/n$ by making use of $M$. This yields the desired approximation ratio of

$$(1 + \varepsilon_0)^{n-1} \le (1 + \varepsilon_0)^n \le 1 + n \cdot \varepsilon_0 = 1 + n \cdot \frac{\varepsilon}{n} = 1 + \varepsilon.$$

We similarly argue for the confidence ratio $\delta_0 := \delta/n$ that yields the desired confidence ratio of $1 - \delta$. That is, the outlined reduction is approximation preserving. What is left is to bound the running time of this recursive procedure. Let $T(n)$ be the running time in question. We have that

$$T(n) \le T(n-1) + t_M(n, 1/\varepsilon_0, 1/\delta_0) + O(1)$$

or $T(n) = O(n \cdot t_M(n, 1/\varepsilon_0, 1/\delta_0)) = O(\text{poly}(n, 1/\varepsilon_0, 1/\delta_0))$. This concludes the proof.

