# OpenReview forum: "Computational Explorations of Total Variation Distance"
_ICLR.cc/2025/Conference — ICLR 2025 Spotlight_

### Official Review · Reviewer_riHb · 2024-10-28

**Soundness:** 4
**Presentation:** 3
**Contribution:** 3
**Rating:** 8
**Confidence:** 3

**Summary:**

This paper studies two fundamental questions about high-dimensional probability distributions.

1. Is there a polynomial-time algorithm which can decide whether two *mixtures of product distributions on the hypercube* are equal to each other?
2. Is there a polynomial-time algorithm which can accurately approximate the total variation distance between two Ising model distributions?

The paper answers (1) affirmatively, giving a very nice linear algebraic algorithm which can test equivalence between mixtures of product distributions on the cube, given access to the parameters of the distributions.
The paper answers (2) negatively, showing that unless NP is contained in RP (that is, unless NP-complete problems have randomized polynomial time algorithms), the total variation distance between Ising models is inapproximable.

Strengths:
The paper studies truly fundamental problems which will be of broad appeal to the ICLR audience.
The results are convincing, and the equivalence-checking algorithm is very nice.

Weaknesses:
These problems are so fundamental that it is a little hard to believe they are not already well studied.
It would be nice if the related work section were expanded.
It would be nice if the algorithm also meant you can test equivalence given samples.

Overall, I recommend accepting the paper to ICLR, on the basis that the results are about a very fundamental set of problems, and the equivalence testing algorithm is very clean.


Questions:
- Does result (1) mean you can also do equivalence testing from samples?
- Is the linear algebraic method you describe similar to any existing equivalence tester in the literature?

**Strengths:**

see above

**Weaknesses:**

see above

**Questions:**

see above

---

> ### Author Response · Authors · 2024-11-22
>
> Dear Reviewer,
>
> Thank you for taking the time to review our paper and your constructive feedback.
>
> We address your questions below.
> 1. This is an interesting question. One approach will be to first learn the distributions from samples and then apply our algorithm. However, such an approach will not give an exact procedure, as learning from samples will necessarily incur some error.
> 2. We are not aware of other equivalence checkers in the literature that use ideas similar to that of our first result.

---

### Official Review · Reviewer_1jgf · 2024-11-02

**Soundness:** 3
**Presentation:** 3
**Contribution:** 2
**Rating:** 6
**Confidence:** 3

**Summary:**

The paper studies some properties about the complexity of computing the Total Variation Distance between distributions. The authors consider  the case of the mixure of product distributions and the case of Ising models. In the first case they show a polytime algo that has access to the marginals and checks the equivalnece of two mixtures. For the second case they show hardness of approximating TVD by an FPRAS  under the hypothesis NP not included in RP.
The two results are sound and well written.
However, the contribution is limited.
I think neither of the two results are very significant alone and together the overall contribution is not very much more.

**Strengths:**

The paper give two new results about computing TVD.
The polynomial time algorithm for TVD of mixtures of product distribution has its main strength in the simplicity and clarity of the approach.
The second result extends the studies on the complexity of dealing with Ising models.

**Weaknesses:**

The paper appears to be a gluing of two minor results with little connection between them.
The second result  builds upoon the Jerrum and Sincler previous analogous study. The first result bears more novelty, although it would not be, in my opinion, sufficient for a paper at ICLR.
I think the main issue is with the specificity (very particular cases) of the two problems solved.

**Questions:**

The paper mentions some open questions but is there any chance that the techniques used, e.g., in the first theorem can be extended to other types of more general distributions? (I personally doubt it)

---

> ### Author Response · Authors · 2024-11-22
>
> Dear Reviewer,
>
> Thank you for taking the time to review our paper and your constructive feedback.
>
> We are not sure whether the techniques used in the first theorem can be extended to other types of more general distributions. In fact, we pose it as an open problem for the mixtures of Bayes nets (with bounded treewidth). Please note that mixtures of product distributions are very general and natural. For instance, it is true that every distribution can be described by some (possibly large) mixture of product distributions. We have added this observation to our appendix (see Appendix A).

---

> > ### Comment · Reviewer_1jgf · 2024-11-26
> >
> > I have read the authors replies to all comments and reviews and I have slightly increased my rating. I still believe the contributi9n is limited.

---

> > > ### Author Response · Authors · 2024-11-26
> > >
> > > Dear Reviewer,
> > >
> > > thank you very much for increasing your rating!

---

### Official Review · Reviewer_5F2F · 2024-11-06

**Soundness:** 4
**Presentation:** 4
**Contribution:** 3
**Rating:** 8
**Confidence:** 4

**Summary:**

The paper has two main contributions: (1) An algorithm for checking equivalence for mixtures of product distributions (2) A hardness reduction for approximating the total variation distance between two Ising models.

The problem of checking equivalence of product distributions is takes the following as input: we are given $w_1,\cdots, w_k, P_1,\cdots,P_k$ and $v_1,\cdots, v_k, Q_1,\cdots,Q_k$ where:
1) Each of $P_1,\cdots,P_k$ and $Q_1,\cdots,Q_k$ is a product distribution over $\Sigma^n$.
2) $w_1,\cdots, w_k$ are the weights for $P_1,\cdots,P_k$ and $v_1,\cdots, v_k$ are the weights for $Q_1,\cdots,Q_k$.
So, the first distribution $P$ is the mixture of $P_1,\cdots,P_k$ each weighted by  $w_1,\cdots, w_k$  and the second distribution $Q$ is the mixture of $Q_1,\cdots,Q_k$ each weighted by  $v_1,\cdots, v_k$. The task is to determine wether the two mixture distributions are the same.

Here is an example (from page 2) an equal mixture of $Bern(1/3)$ and $Bern(2/3)$ equals to the distribution $Bern(1/2)$. This illustrates how a mixture of two product distributions can equal a different product distribution.

The paper gives a deterministic algorithm whose run-time is $O(nk^4 |\Sigma|^4)$ which is polynomial in the input size which equals $kn|\Sigma|$. One idea the algorithm uses is that if the two mixtures $w_1P_1+,\cdots, +P_kw_k$ and $v_1Q_1+,\cdots, +v_k Q_k$ are equal, then (after a normalization) the mixtures $w_1P_1+,\cdots, +P_jw_j$ and $v_1Q_1+,\cdots, +v_j Q_j$ are also equal for every $j$. The algorithm utilizes the idea by iteratively establishing equivalence for $j=1,2..., k$ at every step $j$ establishing equivalence for $j+1$ assuming equivalence for $j$ (or to find some $x$ for which the probabilities of two distributions differ). As the paper shows, such steps can be achieved by keeping track of bases for solutions of certain types of equations.

Ising models are a fundamental family of probability distributions over $\{\pm 1\}^n$. Probability of each $x$ in $\{\pm 1\}^n$ is proportional to $\exp(P(x))$ where $P$ is a degree-2 multilinear polynomial. Ising models are a very well-studied family of distributions modeling systems for which random features have only pairwise interactions (this is because each term in $P$ has at most two variables).

A Fully Polynomial-time Randomized Approximation scheme FPRAS is a randomized algorithm that gives a multiplicative $(1+\epsilon)$-approximation to a desired quantity. Here, this means that one is given a pair of degree-2 polynomials $P_1$ and $P_2$ describing a pair of Ising models, and the goal of the algorithm is to output a multiplicative $(1+\epsilon)$-approximation to the TV distance between the pair of Ising models. The paper shows that no poly-time algorithm can achieve this task (under a basic complexity-theoretic assumption).

The hardness proof proceeds by developing an approximation-preserving reduction to the problem of approximating the partition function of an Ising model, which is known from previous work to be hard to approximate. As an intermediate step, the reduction goes through the problem of approximating the marginal of an Ising-model distribution on one of the coordinates $x_i$ (this is referred to as the problem of approximating the atomic marginal).

**Strengths:**

- Mixtures of products are a fundamental family of probability distributions and checking their equivalence is one of the most basic questions about them.
- The algorithm uses an interesting novel idea of keeping track of bases for the solution spaces of certain equations. This idea might find applications for testing equivalence of other classes of distributions.
- The problem of estimating the total variation distance between two Ising model distributions is quite natural, as Ising models are a very well-studied class of probabilistic models.
- The hardness result only relies on the assumption that NP is not in RP, which is a very mild complexity assumption.

**Weaknesses:**

- The algorithm for mixtures of product distributions can only check whether P=Q exactly. The paper would be stronger if it gave an algorithm for approximating the distance between P and Q.
- The paper rules out FPRAS for TV distance between a pair of Ising models, but it seems that there could still be a constant-factor approximation algorithm, and the paper would be stronger if this question was also addressed (i.e. it was shown that this is also hard, or an algorithm was given).

**Questions:**

- Was a poly-time randomized algorithm for checking equivalence of mixtures of product distributions known previous to this work?
- To the best of your knowledge, what was known from previous work about approximating TV distance between Ising models?
- Do your hardness result tell us anything about hardness of computing an additive approximation for the TV distance between Ising models?
- Does the problem of estimating the TV distance between Ising models remain to be hard if one is given the values of the partition function $Z_1$ and $Z_2$ for each of the Ising models?

---

> ### Author Response · Authors · 2024-11-22
>
> Dear Reviewer,
>
> Thank you for taking the time to review our paper and your constructive feedback.
>
> We will address your questions below.
> 1. We do not know of a randomized algorithm for checking the equivalence of mixtures of product distributions.
> 2. The body of work on TV distance computation is very recent. As we noted in our response to Reviewer Zs6J, the paper
>     >Arnab Bhattacharyya, Sutanu Gayen, Kuldeep S. Meel, and N. V. Vinodchandran. Efficient distance approximation for structured high-dimensional distributions via learning. In Proc. of NeurIPS, 2020
>
>     considers additive approximation of TV distance between ferromagnetic Ising models. However, to the best of our knowledge, the problem of multiplicatively estimating TV distance between Ising models has not been studied before.
> 3. Our hardness result does not imply anything about the hardness of computing an additive approximation for the TV distance between Ising models. In general, the hardness of multiplicative approximation does not say anything about the hardness of additive approximation (for example, the number of satisfying assignments of a CNF formula is hard to multiplicatively estimate, but easy to additively estimate).
> 4. This is a very interesting question which we have not thought about. At this point, even if one is given the values of $Z_1$ and $Z_2$ it is still not clear whether TV distance in general is easy to estimate or not. Note that our proof is based on the hardness of approximation for the partition function (Jerrum and Sinclair, 1993).

---

> > ### Comment · Reviewer_5F2F · 2024-11-25
> >
> > Thank you for your response! My rating remains the same.
> > Please make sure to include your answers above in the final version if accepted.

---

> ### Author Response · Authors · 2024-11-26
>
> Dear Reviewer,
>
> thank you very much!
>
> We will make sure to include our answers above in the final version.

---

### Official Review · Reviewer_Zs6J · 2024-11-08

**Soundness:** 3
**Presentation:** 3
**Contribution:** 2
**Rating:** 6
**Confidence:** 4

**Summary:**

The paper discusses computational aspects of the total variation (TV) distance between probability distributions and has two contributions:

It provides a deterministic polynomial-time algorithm for testing whether two mixtures of product distributions are equivalent, which means the algorithm can decide whether the TV distance between them is zero.

It demonstrates that, unless NP is contained in RP, there cannot exist an efficient algorithm to estimate the TV distance between arbitrary Ising models. This result points out the computational hardness of this problem.

**Strengths:**

The paper addresses important computational questions about the total variation distance, which is fundamental in probability and statistics.

The algorithm for equivalence checking of mixtures of product distributions is new and provides a practical solution to a non-trivial problem.This hardness result bridges complexity theory and statistical measures and provides insight into why certain computational tasks are hard.

The proofs are well written, the results are accessible.

**Weaknesses:**

It would be nicer if the paper could elaborate more on the practical applications of the equivalence checking algorithm with regard to performance on real-world data.

The hardness result could also be pushed further by thinking about the possibility of approximate algorithms with different complexity assumptions.

It would be even more applicable and helpful with more examples or case studies.

**Questions:**

Can the equivalence checking algorithm be extended to other type of mixtures, say Bayesian networks with bounded treewidth?

Is there any hope to soften the hardness assumption in estimating TV distance between Ising models by considering restricted classes of models?

Do the authors have some recommendations for practical algorithms that could approximate the TV distance between Ising models despite the hardness result?

---

> ### Author Response · Authors · 2024-11-22
>
> Dear Reviewer,
>
> Thank you for taking the time to review our paper and your constructive feedback.
>
> We will address your questions below.
> 1. This is a very interesting question. It is not clear to us whether the equivalence-checking algorithm can be extended to other types of mixtures, such as Bayesian networks with bounded treewidth. In fact, we have outlined this question as an open problem (in our Conclusion).
> 2. Regarding our hardness assumption, please note that ${\sf NP} \not \subseteq {\sf RP}$ (that is, SAT does not admit any randomized one-sided-error poly-time algorithm) is a very standard assumption in hardness of approximation results. In fact, the well known result of hardness of approximation of the partition function of the Ising model is under the same assumption (Jerrum and Sinclair, 1993).
> 3. Unfortunately, we do not have in mind any practical approximation algorithm to multiplicatively estimate the TV distance between Ising models. Regarding additive approximations for *ferromagnetic* Ising models, though, one may use the algorithm of the following paper (that is cited in our work):
> >Arnab Bhattacharyya, Sutanu Gayen, Kuldeep S. Meel, and N. V. Vinodchandran. Efficient distance approximation for structured high-dimensional distributions via learning. In Proc. of NeurIPS, 2020.

---

> > ### Comment · Reviewer_Zs6J · 2024-11-26
> >
> > Thanks for your response! My rating remains the same.

---

> > > ### Author Response · Authors · 2024-11-26
> > >
> > > Dear Reviewer,
> > >
> > > thank you very much!

---

### Official Review · Reviewer_VzEw · 2024-11-09

**Soundness:** 3
**Presentation:** 3
**Contribution:** 3
**Rating:** 8
**Confidence:** 3

**Summary:**

This paper addresses the problem of computing the total variation distance (TV-distance) between high-dimensional distributions from a computational standpoint. For distributions with compact descriptions, calculating TV-distance poses significant challenges, as the direct approach incurs exponential time complexity relative to input size. The authors present two main contributions:

(1) They provide a polynomial-time algorithm for determining whether two mixtures of product distributions are identical.

(2) They establish the computational hardness of approximating the TV-distance between two arbitrary Ising models.

The first result is based on a simple and clever algorithm, the second result comes from the standard hardness result of approximating partition functions of Ising models.

Due to the limited time for reviewing, I went through all the proofs and understood the main ideas; however, I did not verify every detailed calculation.

**Strengths:**

The main contribution of this paper is the first result: deciding whether two mixtures of product distributions are the same. Suppose we have $k$ product distributions $P_1, P_2, \ldots, P_k$, where each $P_i$ is an $n$-dimensional product distribution, i.e., $X \sim P_i$ is a vector $(X_1, X_2, \ldots, X_n)$. The paper takes the prefix of $X$, namely $X^{\leq j} = (X_1, X_2, \ldots, X_j)$ for $j \leq n$. This distribution is denoted by $P^{\leq j}_i$. Then, they consider the mixture of $P^{\leq j}_1, P^{\leq j}_2, \ldots, P^{\leq j}_k$, denoted by $P^{\leq j}$. The algorithm decides whether $P^{\leq j}$ and $Q^{\leq j}$ are the same for all $j$. The algorithm is based on induction from $j = 1$ to $j = n$. The base case is trivial. The difficult part is that for $P^{\leq j}$ and $Q^{\leq j}$, the support of the distribution can be as large as $\exp(\Omega(j))$. To reduce the computational cost, the algorithm finds a "sketch" of the two distributions. One needs to check whether $P^{\leq j}(x) = Q^{\leq j}(x)$ for exponentially many $x \in \Sigma^{j}$. For each $x$, the algorithm views $P^{\leq j}(x) = Q^{\leq j}(x)$ as a linear equation. Instead of checking an exponential number of linear equations, the algorithm finds a basis of the linear system, and the size of the basis is $\text{poly}(n)$. Then the algorithm only need to check the equations in the basis.

Overall, the algorithm and the definition of $P^{\leq j},Q^{\leq j}$ are simple and clever, and I think deciding whether two mixtures of product distributions are the same is a basic problem in statistics.

**Weaknesses:**

The hardness result follows from standard results. Proposition 6 provides a self-reduction for the Ising model, which is used in the standard counting-to-sampling reduction. Therefore, the proof of Proposition 6 could be omitted. Proposition 8 essentially states that one can fix the value of a vertex $v$ by adjusting the function $h(v)$, allowing the TV distance to encode the marginal distribution.

The relationship between Theorem 1 and Theorem 2 is not very strong, as they pertain to different models.

**Questions:**

I think the logic in lines 181–184 is incorrect, though the result is ultimately correct. The logic appears to use the implication $ A \implies B $ to conclude $ \neg A \implies \neg B $, which is not valid. The result holds, however, because $ P = \sum_{i=1}^k w_i P_i^{\leq n} $ and $ Q = \sum_{i=1}^k v_i Q_i^{\leq n} $, so if $ P \neq Q $, then $ \sum_{i=1}^k w_i P_i^{\leq j} \neq \sum_{i=1}^k v_i Q_i^{\leq j} $ for at least $ j = n $.

For line 451, it should be $ \delta x_0 x_1 \to \delta x_0 x_k $. Additionally, some of the calculations in lines 441–458 are straightforward, so a few lines could be removed to streamline the presentation.

In the proof of Proposition 7, when setting parameters, it would be helpful to specify how small the parameter $ \eta_0 $ ( and $h_0,\delta$) needs to be to ensure that the relative error of the marginal probability remains smaller than $ O(\epsilon / n) $. Could you clarify why machine precision is considered here? Would $ \mathrm{poly}(N / \epsilon) $ bits suffice to represent $ \eta_0 $, where $ N $ denotes the input size?

---

> ### Author Response · Authors · 2024-11-22
>
> Dear Reviewer,
>
> Thank you for taking the time to review our paper and your constructive feedback.
>
> We will address your comments below.
> 1. We took care of the logical error and typo pointed out. Thank you!
> 2. Regarding the remark about Proposition 6, we have moved the proof of Proposition 6 to the appendix (see Appendix B).
> 3. Regarding your question about Proposition 7, we have updated the proof to include more details about the choice of the associated parameters $h_0$ and $\delta$. See the Lines 449 -- 485 of the revision.

---

> > ### Comment · Reviewer_VzEw · 2024-11-24
> >
> > Thank you for your reply. For the logic error, in your revision, you added a "if and only if" relation, which is necessary for your algorithm. I think it is true, but it would be good to have a few sentences to explain two directions, because it is important to the correctness of your algorithm.

---

> > > ### Author Response · Authors · 2024-11-24
> > >
> > > Dear Reviewer,
> > >
> > > thank you very much for your follow-up comment!
> > >
> > > Please feel free to have a look at our new revision, whereby we implement your proposal.

---

> > > > ### Comment · Reviewer_VzEw · 2024-11-26
> > > >
> > > > Thank you for your reply. My rating remains the same.

---

> > > > > ### Author Response · Authors · 2024-11-26
> > > > >
> > > > > Dear Reviewer,
> > > > >
> > > > > thank you very much!

---

### Comment · Area_Chair_CiFW · 2024-11-26
**Response**

Dear Reviewers,

The authors have provided their rebuttal to your questions/comments. It will be very helpful if you can take a look at their responses and provide any further comments/updated review, if you have not already done so.

Thanks!

---

### Meta-Review · Area_Chair_CiFW · 2024-12-20

**Metareview:**

The total variation distance between multivariate distributions is a basic function that appear often in statistics and information theory. Computing TV distance between even simple product distributions can be difficult to understand. This paper first gives an algorithm for testing, and then provides a conditional lower bound for Ising models.

Weakness: appeal to ICLR audience may be less

Reviewers are positive about the paper and gave high scores. I recommend acceptance.

**Additional Comments On Reviewer Discussion:**

The discussions with authors seem to be precise and straightforward.

---

### Decision · Program_Chairs · 2025-01-22

Accept (Spotlight)